# Der f 38 Is a Novel TLR4-Binding Allergen Related to Allergy Pathogenesis from *Dermatophagoides farinae*

**DOI:** 10.3390/ijms22168440

**Published:** 2021-08-05

**Authors:** Geunyeong Kim, Minhwa Hong, Ayesha Kashif, Yujin Hong, Beom-Seok Park, Ji-Young Mun, Hyosun Choi, Ji-Sook Lee, Eun-Ju Yang, Ran-Sook Woo, Soo-Jin Lee, Minseo Yang, In-Sik Kim

**Affiliations:** 1Department of Senior Healthcare, Eulji University, Uijeongbu 11759, Korea; ns9017@naver.com (G.K.); mystzr@naver.com (M.H.); ashabbasi@gmail.com (A.K.); doremi0507@naver.com (Y.H.); 2Department of Biomedical Laboratory Science, College of Health Science, Eulji University, Seongnam 13135, Korea; bspark74@eulji.ac.kr; 3Neural Circuit Research Group, Korea Brain Research Institute, Daegu 41068, Korea; mjy1026@gmail.com; 4Nanobioimaging Center, National Instrumentation Center for Environmental Management, Seoul National University, Seoul 08826, Korea; hyokchoi0123@gmail.com; 5Department of Clinical Laboratory Science, Wonkwang Health Science University, Iksan 54538, Korea; jslee1216@wu.ac.kr; 6Department of Clinical Laboratory Science, Daegu Haany University, Gyeongsan 38610, Korea; ejyang@dhu.ac.kr; 7Department of Anatomy and Neuroscience, Eulji University School of Medicine, Daejeon 34824, Korea; rswoo@eulji.ac.kr; 8Department of Pediatrics, Eulji University School of Medicine, Daejeon 34824, Korea; sjlee@eulji.ac.kr; 9Department of Biomedical Laboratory Science, College of Health Science, Eulji University, Uijeongbu 11759, Korea; yangmin236@naver.com

**Keywords:** house dust mite, allergy, TLR4, Der f 38

## Abstract

It is difficult to treat allergic diseases including asthma completely because its pathogenesis remains unclear. House dust mite (HDM) is a critical allergen and Toll-like receptor (TLR) 4 is a member of the toll-like receptor family, which plays an important role in allergic diseases. The purpose of this study was to characterize a novel allergen, Der f 38 binding to TLR4, and unveil its role as an inducer of allergy. Der f 38 expression was detected in the body and feces of *Dermatophagoides farinae* (DF). Electron microscopy revealed that it was located in the granule layer, the epithelium layer, and microvilli of the posterior midgut. The skin prick test showed that 60% of allergic subjects were Der f 38-positive. Der f 38 enhanced surface 203c expression in basophils of Der f 38-positive allergic subjects. By analysis of the model structure of Der p 38, the expected epitope sites are exposed on the exterior side. In animal experiments, Der f 38 triggered an infiltration of inflammatory cells. Intranasal (IN) administration of Der f 38 increased neutrophils in the lung. Intraperitoneal (IP) and IN injections of Der f 38 induced both eosinophils and neutrophils. Increased total IgE level and histopathological features were found in BALB/c mice treated with Der f 38 by IP and IN injections. TLR4 knockout (KO) BALB/c mice exhibited less inflammation and IgE level in the sera compared to wild type (WT) mice. Der f 38 directly binds to TLR4 using biolayer interferometry. Der f 38 suppressed the apoptosis of neutrophils and eosinophils by downregulating proteins in the proapoptotic pathway including caspase 9, caspase 3, and BAX and upregulating proteins in the anti-apoptotic pathway including BCL-2 and MCL-1. These findings might shed light on the pathogenic mechanisms of allergy to HDM.

## 1. Introduction

House dust mites (HDMs) play pivotal roles in inducing allergic diseases [1,2]. *Dermatophagoides farinae* (DF) is a major HDM that contains active allergic components such as arginine kinase, chitinase, cysteine protease, serine protease, fatty acid-binding protein, and peritrophin-like protein related to allergic responses [3]. Until now, a total of 39 allergens of DF (Der f 1-Der f 39) have been registered in the WHO/IUIS Allergen Nomenclature Sub-committee. These proteins can activate immune responses and lead to allergic sensitization [3,4]. Atopic subjects with exposure to HDM allergens show high levels of total IgE and allergen-specific IgE in their sera compared to non-atopic allergic patients. They also have a high risk for developing incapacitating disease manifestations such as asthma, allergic rhinitis, and atopic dermatitis [5,6,7]. 

Toll-like receptors (TLRs), protease-activated receptors (PARs), and NOD-like receptors (NORs) can be activated by HDM allergens, thus promoting allergic inflammation and immune deviation [8]. In particular, TLR4 is an essential receptor in HDM-mediated allergy [9]. TLR4 activation leads to the formation of the MyD88 complex. This induces the activation of TRAF6 and TAK1, and finally promotes NF-κB translocation.

As group 2 critical mite allergens, Der f 2 and Der p 2 have structural homologies with MD-2 that can bind to TLR4. Thus, Der f 2 and Der p 2 are called MD2-mimic proteins [10]. Der p 2 carries its signal to drive airway T helper (Th) 2 inflammation via TLR4 [11]. On the other hand, Der p 2 shows remarkable skin inflammation in TLR4 knockout (KO) mice, indicating that TLR4 has different roles depending on individual allergic disease [12]. Although Der f 35 has recently been found to be homologous to Der f 2 and MD2, its allergic mechanism remains unknown [13]. Additionally, Der f 33, a novel protein, has allergenicity alone [14], although the concise mechanism and signal transduction involved in its allergenicity remain unknown.

As a novel protein of DF binding to TLR4, Der f 38 has been recently found and approved by the WHO/IUIS Allergen Nomenclature Sub-committee. The aim of this study was to examine allergic responses to Der f 38 and determine the mechanisms involved in such allergic responses. 

## 2. Results

### 2.1. Identification of Der f 38 in DF

We have recently identified Der p 38 in *Dermatophagoides pteronyssinus* (DP). It was registered by the WHO/IUIS Allergen Nomenclature Sub-committee. DF whole genome was examined and a novel protein homologous to Der p 38 was identified. As shown in Figure 1A, Der f 38 shared homologies with Der p 38 (80% identities for full sequences and 82% identities after excluding expected signal peptides (1–20 aa)). A recombinant Der f 38 protein was generated as mentioned in the Materials and Methods section (Figure 1A). Antibodies against Der f 38 were produced by the sensitization of recombinant Der f 38 that enabled the separation of native Der f 38 from the DF extract. Eluted native Der f 38 was electrophoresed and subjected to silver staining (Figure 1C). It was confirmed by Q-TOF using chymotrypsin (Appendix A). In addition, Der f 38 was detected in DF body and feces based on Western blotting (Figure 1D). Because the allergen detected in fecal pellets was located in the digestive system [15,16], we examined whether Der f 38 was present in this system, particularly posterior midgut, by electron microscopy. Results showed that Der f 38 was widespread from the granule layer to microvilli in the posterior midgut. However, there was no detection of Der f 38 between the cuticle and the granule layer (Figure 1E and Appendix A).

### 2.2. Allergenicity of Der f 38 

We next conducted a skin prick test to examine the sensitization of Der f 38 in allergic subjects. A positive reaction to Der f 38 was found in 60% of allergic subjects (Table 1). Additionally, we performed a dot blot assay. The frequency of IgE reactivity to Der f 38 was 75% (15/20) in allergic subjects (Figure 2A). Basophil activation due to allergen reflects its allergenicity. Therefore, we investigated the alteration of CD203c expression as a basophil activation marker. As shown in Figure 2B, CD203c expression in basophils from Der f 38-positive allergic subjects was significantly increased by stimulation with Der f 38 in a dose-dependent manner. In contrast, there was no change in CD203 expression in normal or Der f 38-negative allergic subjects. To unveil epitope sites to induce allergenicity, we first analyzed the structure of Der f 38 using Pymol software. Der f 38 has three α helices and four β sheets (Figure 1A and Figure 2C). Based on bioinformatics and a protein database, we predicted one signal peptide (aa 1–20) and two epitope regions [aa 46–62 (GKSKG IGEGANIVGF) and aa 121–137(GRMVNAPKPG TKVREEN)]. These epitopes are located at the exterior of Der f 38.

### 2.3. Der f 38 Induces Airway Inflammation and Increases Total and Der f 38-Specific IgE via TLR4 in Mice

To determine the effect of Der f 38 in allergies in vivo, we examined the inflammatory status and mechanism of Der f 38 in a mouse model. IN and IN/IP injection of the allergen could lead to different clinical features. Therefore, we performed both IN and IN/IP injections of Der f 38 and evaluated histopathological and serological manifestations in the lungs and sera of mice. As shown in Figure 3B, IN administration of Der f 38 enhanced neutrophil infiltration as a general effect of an allergen. The group with IN/IP exposure to Der f 38 (Der f 38/Der f 38) showed an increase of total cell count, similar to the number of inflammatory cells in the DF/DF group, indicating that Der f 38 induced an inflammatory response in the lung (Figure 3A–D). Notably, Der f 38 induced both eosinophils and neutrophils. In contrast, the DF extract increased eosinophils without increasing neutrophils (Figure 3B). This phenomenon was observed when lung tissues were stained with H&E and immunologically stained with anti-Ly6G, Eosinophil peroxidase, and tryptase antibodies (Figure 3C,D). Mucus secretion and a number of eosinophils, neutrophils, and mast cells were boosted by IN/IP injection of Der f 38. Der f 38-specific IgE and total IgE levels were increased by Der f 38 IN/IP treatment, but not by IN treatment (Figure 3E,F). 

TLR4 is a critical receptor in HDM-mediated allergy. Thus, we investigated whether Der f 38 could elicit inflammatory effects via TLR4. *TLR4* KO mice were used in this experiment. Ablation of TLR4 suppressed the increase of total cell count and inhibited the infiltration of eosinophils and neutrophils due to Der f 38 (Figure 4A,B). After IN or IN/IP injection of Der f 38 to *TLR4* KO mice, increases of mucus production and leukocyte infiltration were not seen compared to the WT group (Figure 4C,D). Both total IgE and Der f 38-specific IgE levels were not altered or slightly increased but not significantly increased in *TLR4* KO mice (Figure 4E,F). These results led us to determine whether Der f 38 could directly interact with TLR4. Der f 38 binds to TLR4 (*K_D_* = 74 nM), although the affinity of Der f 38 for TLR4 is lower than that of MD2, a molecule showing the highest affinity for TLR4 (*K_D_* = 1.56 nM) (Figure 5).

### 2.4. Der f 38 Suppresses Constitutive Apoptosis of Neutrophils and Eosinophils

Since numbers of neutrophils and eosinophils in the lungs of WT mice were increased after treatment with Der f 38 (Figure 3B,D), Der f 38 might be an inducer of both neutrophils and eosinophils. We then investigated the mechanism involved in the effect of Der f 28. Der f 38 delayed apoptosis of allergic and normal neutrophils in a dose-dependent manner (Figure 6A, and Appendix A). Der f 38 also blocked eosinophil apoptosis in 25% of allergic subjects (2/8) (Figure 3B). This inhibition due to Der f 38 was associated with TLR4 (Figure 6D). As shown in Figure 6E,F, Der f 38 delayed the activation of caspase 9 and caspase 3 in a time-dependent manner during spontaneous apoptosis. Additionally, Der f 38 inhibited the decrease of expression of anti-apoptotic proteins such as MCL-1 and BCL-2. It also inhibited the increase of BAX, a pro-apoptotic protein (Figure 6F).

## 3. Discussion

After the identification of critical allergens such as Der f 1 and Der p 1, a variety of HDM allergens have been found by proteomics using serum IgE of allergic patients and genomic strategies using HDM [17,18,19,20]. Der f 38 was identified in the DF whole genome by alignment with the Der p 38 sequence. Previous studies have recently reported IgE binding efficiencies of DF-related allergens. By ELISA, recombinant Der f 35 shows 51.4% IgE-binding efficiency, lower than native Der f 35 (binding efficiency of 77.5%) [13]. Native Der f 2 shows 65% positive reaction, lower than recombinant Der f 2 (94.2%) [13]. Post-translation of the allergen and structure change might affect the detection of allergen-specific IgE, although native or recombinant protein is more efficient in binding to IgE remains unclear [21,22]. By the skin prick test, Der f 33 shows a 23.5% positive reaction in allergic subjects and Der f 21 was positive in 42.9% of allergic subjects [14,23]. Several methods such as the skin prick test, ELISA, and the dot blot are widely used for detecting allergen-specific IgE. Allergen sensitivity might be altered by different detection methods and characteristics and the status of allergic subjects [24,25,26,27]. The sensitivity of Der f 38 to allergic patients is 60% by the skin prick test and 75% by a dot blot assay (Table 1 and Figure 2A). Der f 38 can be considered as an allergen showing either moderate or high IgE-binding frequency in allergic subjects, indicating that it is closely related to the pathogenesis of allergy.

Before unveiling the specific mechanism due to Der f 38, we focused on the extraordinary effect of Der f 38 that promoted the movement of both neutrophils and eosinophils (Figure 3C,D). IN/IP injection of the HDM extract usually increases eosinophil infiltration in mice while IN administration markedly enhances the number of neutrophils in the lungs [28,29]. In our experiment, IN/IP treatment with the DF extract induced eosinophil infiltration, in contrast with the effect of Der f 38. Such a differential function of Der f 38 led us to investigate apoptosis of neutrophils and eosinophils, an overarching step in the regulation of inflammation [30,31,32]. Der f 38 directly inhibited apoptosis of neutrophils and eosinophils of normal and allergic subjects (Figure 6A and Appendix A). However, Der f 38 did not affect the apoptosis of all eosinophils. Der f 38 suppressed eosinophil apoptosis in 25% (2/8) of allergic subjects and 33.3% (1/3) of normal subjects. It did not significantly suppress all eosinophils of normal or allergic subjects (Figure 6B and Appendix A). These results indicate that the anti-apoptotic effect of Der f 38 on eosinophils might depend on unknown characteristics of individual normal and allergic subjects compared to its effect on neutrophils. There was no difference in the anti-apoptotic effects of Der f 38 on neutrophils and eosinophils between Der f 38-positive and Der f 38-negative asthmatic subjects (Appendix A). Although TLR4 has shown an anti-allergic effect in a few papers, it is considered to be an orchestrating factor related to the onset and alleviation of allergy [33,34,35,36]. Our results demonstrated that TLR4 was a key receptor of allergic effects due to Der f 38 in this study (Figure 4 and Figure 5). In addition, the pathogenic roles of macrophage subsets and innate lymphoid cells (ILCs) are closely associated with allergic diseases [37,38,39,40]. Macrophages can be classified into either M1 or M2 phenotypes. The polarization process of macrophages is involved in asthma development. ILC2 and ILC3 function as activators in asthma pathogenesis. Further studies on Der f 38 need to focus on the specific mechanisms including macrophage polarization and ILC subsets in allergic diseases 

Recent research works are making progress in the development of immunotherapy for allergy using a specific mite allergen. Finding the structure and epitope of the allergen is fundamental to conduct specific immunotherapy. Structures of allergens have been analyzed and the target epitope has been used in the production of antibodies against the target epitope [41,42]. A tablet made of the DP extract has also been applied in allergy treatment. There is no relationship between the clinical efficacy of immunotherapy and sensitization to each mite allergen [43]. Der f 38 might be useful for immunotherapy using its peptide, the chemically modified Der f 38 [44]. Further studies are needed to unveil a more detailed structure and appropriate epitopes in the near future.

## 4. Materials and Methods

### 4.1. Reagents

DF extract was purchased from Cosmo Bio (Tokyo, Japan). DF feces were obtained from INDOOR Biotechnologies (Charlottesville, VA, USA). Adult DF mites were obtained from the Korea National Arthropods of Medical Importance Resource Bank (Seoul, Korea). Recombinant TLR4 and MD2 proteins were obtained from R&D Systems (Minneapolis, MN, USA). RPMI 1640, fetal bovine serum, and the TRIzol reagent were purchased from Life Technologies Inc. (Gaithersburg, MD, USA). Antibodies against Ly6G and tryptase were obtained from Abcam (Cambridge, UK). The antibody against eosinophil peroxidase was purchased from Bioss Antibodies (Woburn, MA, USA). The anti-Bcl-2 antibody was obtained from BD Biosciences (Heidelberg, Germany). Anti-BAX, anti-MCL-1, anti-cleaved caspase 9, anti-cleaved caspase 3, anti-rabbit IgG-HRP, and anti-mouse IgG-HRP antibodies were obtained from Cell Signaling Technology (Danvers, MA, USA). The anti-ERK2 antibody was obtained from Santa Cruz Biotechnology (Santa Cruz, CA, USA).

### 4.2. Production of Recombinant Der f 38

The TRIzol reagent (Life Technologies) was used to extract total RNA from live DF. RNeasy Mini Kit (Qiagen, Hilden, Germany) was used to purify extracted RNA. Purified RNA was subsequently treated with DNase I (New England Biolabs, Ipswich, MA, USA). Finally, cDNA was synthesized from DNase I treated RNA using an iScript cDNA synthesis kit (Bio-Rad, Hercules, CA, USA). Mature Der f 38 was amplified using PCR and cloned into pETDuet-1 (Merck Millipore, Darmstadt, Germany) for expression. His-tagged Der f 38 was purified using a nickel column (Merck Millipore) followed by affinity chromatography using a Superdex 200 column attached to an ÄKTA FPLC system (GE Healthcare, Chicago, IL, USA). A ToxinSensor Chromogenic LAL Endotoxin Assay Kit (GenScript, Piscataway, NJ, USA) was used to assess the endotoxin level. The removal of endotoxin was performed using a ToxinEraser Endotoxin Removal Kit (GenScript). Purity > 95% was confirmed by sodium dodecyl sulfide (SDS)-polyacrylamide gel electrophoresis (PAGE).

### 4.3. Detection of Native Der f 38 in DF Extract

The polyclonal antibody against Der f 38 was produced with a general procedure including first sensitization and secondary boosting in rabbits. The anti-Der f 38 antibody was attached to a column packed with DEAE sepharose fast-flow resin. The DF extract was then added to the column. The column was then washed with PBS and eluted with 1M NaCl. The eluted protein was separated by SDS-PAGE followed by silver staining.

### 4.4. Block Preparation for Transmission Electron Microscopy (TEM) and Immuno-Gold Staining for Der f 38 Labeling

Adult DF mites were fixed with 2.5% glutaraldehyde and 2% paraformaldehyde in 0.1 M cacodylate solution (pH 7.0) for 3 days at 4 °C and post-fixed with 2% osmium tetroxide for 48 h at 4 °C. The block was stained with 1% TCH and 2% uranyl acetate solution and then dehydrated with a graded ethanol series. Block sections were also stained using the NCMIR method. Stained samples were then embedded into an epoxy medium (EMS, Delaware, OH, USA). After incubating with primary anti-Der f 38 polyclonal antibody diluted with 0.1% blocking solution, blocks were washed several times and incubated with secondary anti-rabbit antibody conjugated to 10 nm gold particles (Sigma-Aldrich Korea, Seoul, Korea) at room temperature for 1 h. Gold particles were then observed at 200 kV using a Tecnai G2 (FEI, Hillsboro, OR, USA).

### 4.5. Normal and Allergic Subjects

Twenty allergic subjects with allergic rhinitis (AR) [12], asthma [2], atopic dermatitis (AD) [4], AR/asthma [1], or AR/AD [1] were recruited from Eulji University (Table 1). Allergic status was based on history and the presence of house dust mite (HDM)-positive skin prick test results. Twenty healthy subjects were recruited as normal controls. These normal subjects had no prior history of allergy with any other manifestations of other diseases. Normal and allergic subjects were currently not on any medication. This study was approved by the Institutional Review Board of Eulji University. All participants in this study provided written informed consent. 

### 4.6. Basophil Activation Assay

Allergen-specific basophil activation was analyzed by evaluating the expression of CD203c, a basophil activation marker, and IgE using flow cytometry. Granulocytes were isolated from heparinized peripheral blood of healthy and allergic subjects using Ficoll–Hypaque gradient centrifugation. Erythrocytes and neutrophils were removed by RBC lysis buffer and CD16 microbead magnetic cell-sorting kit (Miltenyi Biotec, Bergisch Gladbach, Germany), respectively. Isolated cells were collected and washed with PBS buffer. Cells containing eosinophils and basophils were stimulated with different concentrations (0.1, 1, or 10 μg/mL) of Der f 38. PBS was used as a negative stimulator. Cells were then incubated at 37 °C for 30 min. Processed cells were incubated on ice for 5 min, washed with PBS, and incubated with PE-conjugated anti-human IgE and FITC-conjugated CD203c (Biolegend, San Diego, CA, USA) for 20 min on ice. Following washing with the PBS buffer, stained cells were analyzed on an RF-500 (Sysmex Corporation, Kobe, Japan). Flow cytometric analysis was performed with RF-500 software (Sysmex Corporation, Kobe, Japan), which detected the intensity of CD203c in basophils out of IgE-positive cells. 

### 4.7. Skin Prick Test

Conventional skin prick tests were conducted according to the instructions of a SoluprickR test. Test reagents included DP, DF (ALK Horsholm, Denmark), and recombinant Der f 38 protein (10 μg/mL). Histamine was used as a positive control. Each subject’s skin was pricked with a lancet. Each allergen was then dropped to the pricked skin at 2 cm intervals to avoid overlapping reactions and false positivity. The test result was measured 15–20 min after the application. A positive result was defined as a wheal ≥ 3 mm in diameter. 

### 4.8. Dot Blot Assay

A dot-blot assay was performed to determine serum IgE level. Briefly, protein samples (0.5 μg each) were dotted onto nitrocellulose strips followed by blocking with 3% BSA in TBS-T. These strips were then incubated with normal or patient sera at a 1:10 dilution overnight at 4 °C. Bound IgE was detected with a secondary mouse anti-human IgE antibody and visualized with a chemiluminescence detection system (Thermo Scientific, Waltham, MA, USA).

### 4.9. Modeling of the Tertiary Structure of Der f 38

Data are presented as mean ± standard error (SD) and represent the average of three independent experiments. The SPSS statistical software package (Version 18.0, Armonk, NY, USA) was used for the analysis of variance (ANOVA), as appropriate. Additionally, the differences between the groups were compared with the one-way ANOVA, followed by Scheffe and Dunnett T3 methods. The results with *p*-values < 0.05 were considered statistically significant.

### 4.10. Asthma Induction by Der f 38 Administration in Mice 

Six-week-old female WT and TLR4 knockout (KO) BALB/c mice were maintained in a specific pathogen-free (SPF) facility. These mice were assigned into four groups (*n* = 5 mice per group): (1) PBS/PBS group, where mice were intraperitoneally (IP) injected with PBS (50 μL) on days 1 and 14. These mice were intranasally (IN) administered PBS (50 μL) from days 21 to 27 after the second sensitization; (2) -/Der f 38 group, where mice were IN injected with Der f 38 (50 μg/50 μL) for one week without IP injection; (3) Der f 38/Der f 38 group, where mice were IP injected with Der f 38 (100 μg/100 μL) on days 1 and 14. These mice were intranasally administered Der f 38 or (50 μg/50 μL) from days 21 to 27 after the second sensitization; (4) DF/DF group, where mice were IP injected with DF (100 μg/100 μL) on days 1 and 14. These mice were intranasally administered DF (50 μg/50 μL) from days 21 to 27 after the second sensitization.

The PBS-treated group was used as a negative control. These groups were coined according to stimulators administered by IP/IN injection. All animal experiments performed in this study were approved by the Institutional Animal Care and Use Committee of Eulji University, Korea.

### 4.11. Collection of Bronchoalveolar Lavage Fluid (BALF) and Serum 

BALF was collected by lung lavage via the trachea with 1 mL of PBS five times. Blood was collected by heart puncture. BALF and blood were centrifuged. Supernatants were collected and stored at −70 °C. Cells in the BALF were resuspended in 100 μL of PBS for total cell count and differential counts. Total cell numbers were counted using a Neubauer hemocytometer. Cells suspended in PBS were attached to a slide by cytospinning and stained with a Diff Quick kit (Sysmex Corporation). The ratio of leukocytes, including eosinophil, neutrophil, macrophage, and lymphocyte, was presented as a percentage. 

### 4.12. Detection of Total IgE and Der f 38-Spexific IgE in the Sera 

To measure the concentration of IgE binding to recombinant Der f 38 protein, plates were coated with 1 μg/mL of Der f 38 in carbonate-buffered solution (15 mM Na2CO3 and 35 mM NaHCO3, pH 9.5) per well at 4 °C overnight. To prevent non-specific binding, plates were blocked with 3% BSA solution at 37 °C for 1 h and then incubated with 1:5 diluted serum at 37 °C for 1 h. After incubation with biotin-conjugated goat anti-mouse IgE (1:2000) and SAv-HRP reagent for 1 h at 37 °C, the reaction was stopped by adding 2M H2SO4 solution. Total IgE levels in the sera of mice were evaluated using a Mouse IgE ELISA set (BD Biosciences). OD values were measured at 450 nm with an ELx808 absorbance microplate reader (BioTek, Winooski, VT, USA). 

### 4.13. Histological Analysis

Experimental animals were euthanized. Lung tissues were separated and fixed in formalin solution. These fixed tissues were embedded in paraffin and cut into 5-μm sections using a microtome (Leica Microsystems, Wetzlar, Germany). Blocks were deparaffinized and stained with hematoxylin-eosin or periodic acid Schiff (PAS) stain (Sigma-Aldrich Korea) to detect inflammation or mucus production. In immunohistochemical staining, 3 μm-thick sections were laid on SuperfrostPlus microscope slides (Fisher Scientific). The Vectastain elite ABC HRP kit (Vector Labs) was used as a 3,3′-diaminobenzidine (DAB) chromogen for detecting antibodies. Sections were deparaffinized by xylene. A solution of proteinase K was added and incubated for 30 min for antigen retrieval. Sections were then treated with 0.3% H2O2 solution for 30–40 min. Slides were blocked with a blocking solution and subsequently incubated with primary antibodies and secondary antibodies. After incubating with AB and DAB reagents, specimens were counterstained with H&E and examined under a light microscope (Leica Microsystems).

### 4.14. Biolayer Inteferometry 

The binding of Der f 38 or MD2 with TLR4 was evaluated by biolayer interferometry (BLI) using an Octet Qke machine (Fortebio, Fremont, CA, USA). Recombinant TLR4 (1 μg/mL) was immobilized in phosphate-buffered saline (PBS) by amine-coupling on an AR2G sensor chip. To prevent non-specific binding, a blocking step was performed using 1M ethanolamine (pH 8.5). The biosensor was equilibrated in an assay buffer. Sensors were dipped in 2-fold serial dilutions of Der f 38 (6400 nM, 3200 nM, 1600 nM, 800 nM, 400 nM, and blank control) for 300 s during the association phase. Dissociation was then proceeded for 1200 s in the assay buffer. BLI data acquisition software 9.0.0.26 and data analysis software 9.0.0.10 (Fortebio) were used to produce and evaluate the data.

### 4.15. Isolation of Neutrophils and Eosinophils and Western Blotting 

Human neutrophils and eosinophils were isolated from heparinized peripheral blood samples of normal and allergic subjects using Ficoll–Hypaque gradient centrifugation and a CD16 microbeads magnetic cell sorting kit (Miltenyi Biotec, Bergisch Gladbach, Germany). The purities of both cells were greater than 97%. After stimulation with Der f 38, cell lysates were collected and loaded on gels. SDS-PAGE and Western blotting were performed as described previously [45,46]

### 4.16. Detection of Apoptosis 

An annexin V–fluorescein isothiocyanate (FITC) apoptosis detection kit (BD Biosciences) was used for apoptosis evaluation as described previously [47]. Apoptotic cells were analyzed using a FACSCalibur flow cytometer (BD bioscience) and reported as the percentage of cells showing annexin V+/propidium iodide (PI)- and annexin V+/PI+. 

### 4.17. Statistical Analysis

Data are expressed as means ± SD. Statistical differences were analyzed using a paired t-test for two-group comparisons and one-way ANOVA for comparison of more than two groups. All analyses were conducted using the SPSS statistical software package (Chicago, IL, USA). Statistical significance was considered at *p* < 0.05.

## 5. Conclusions

Der f 38 has allergenicity in allergic subjects and strongly induces the infiltration of eosinophils and neutrophils in asthma-like mice. Der f 38 is a potential allergen in allergy pathogenesis. 

## Figures and Tables

**Figure 1 ijms-22-08440-f001:**
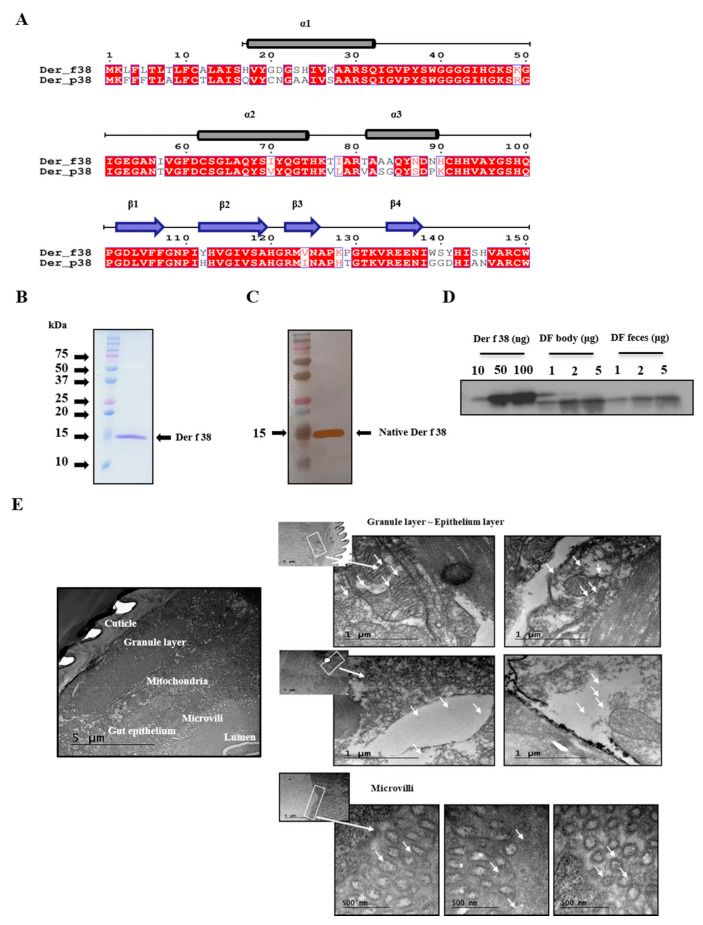
Identification of Der f 38 in DF intestine. (**A**) Der f 38 amino acid sequence is aligned with Der p 38 protein. Red box shows strict identity. Blue frame shows similarity across groups. Red character indicates similarity in a group. (**B**) Der f 38 cDNA was amplified by PCR and cloned to a vector. After transformation of the vector into E. coli, recombinant Der f 38. protein was produced. The Der f 38 protein was separated by SDS-PAGE and stained with Coomassie brilliant blue. (**C**) The native Der f 38 protein in DF extract was separated by a column attached to anti-Der f 38 polyclonal antibodies. Gel loaded with the protein was electrophorized and then subjected to silver staining. (**D**) Recombinant Der f 38, DF body, and DF feces at the indicated concentrations were analyzed by Western blotting using antibodies against Der f 38. (**E**) The presence of Der f 38 in the posterior midgut of DF was detected by electron micrograph.

**Figure 2 ijms-22-08440-f002:**
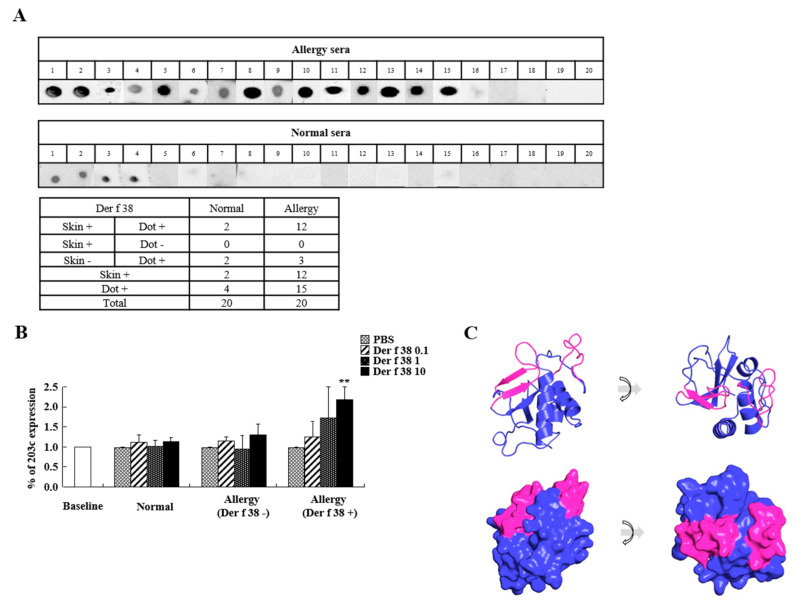
IgE reactivity and basophil activation of Der f 38. (**A**) A dot blot assay was performed to detect IgE reactivity of Der f 38 using sera of normal and allergic subjects (upper panel). Comparison of skin prick test results with dot blot assay results (lower panel). (**B**) Granulocytes were isolated from normal and allergic subjects and treated with PBS or Der f 38 at the indicated concentrations (μg/mL). CD203c expression in basophils was evaluated using flow cytometry. Data are presented as the mean ± S.D. ** *p* < 0.01, significant differences between PBS-treated and Der f 38-treated groups. (**C**) Three-dimensional structure of Der f 38 was predicted with the swiss-model (https://swissmodel.expasy.org/repository/uniprot/A0A6B9VTT9, accessed on 15 July 2020) based on Der p 38 sequence and presented with Pymol software (upper panel: Cartoon; lower panel: Surface). Predicted epitope sites are shown in pink color.

**Figure 3 ijms-22-08440-f003:**
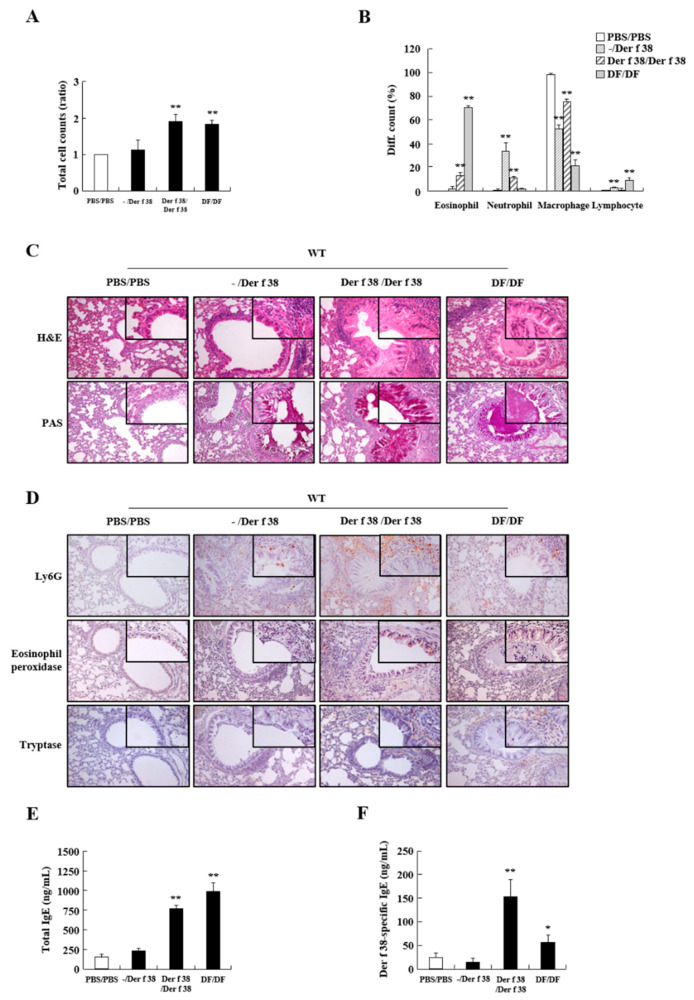
Der f 38 triggers infiltration of inflammatory cells into the lung and mucin hypersecretion in WT mice. (**A**,**B**) Total cell (**A**) and differential cell counts (**B**) in BALF of BALB/c mice after the IN or/and IP administration of PBS, Der f 38, or DF extract. (**C**,**D**) Lung tissues from mice were stained with hematoxylin and eosin (H&E) and periodic acid-Schiff (PAS) (**C**) or stained with antibodies against Ly6G, eosinophil peroxidase, and tryptase (**D**). (**E**,**F**) Total IgE (**E**) and Der f 38-specific IgE (**F**) levels in mice sera were evaluated. Data are presented as the mean ± S.D. * *p* < 0.05; ** *p* < 0.01, significant differences between control and stimulator-treated groups. Magnification, ×200 (**C**,**D**).

**Figure 4 ijms-22-08440-f004:**
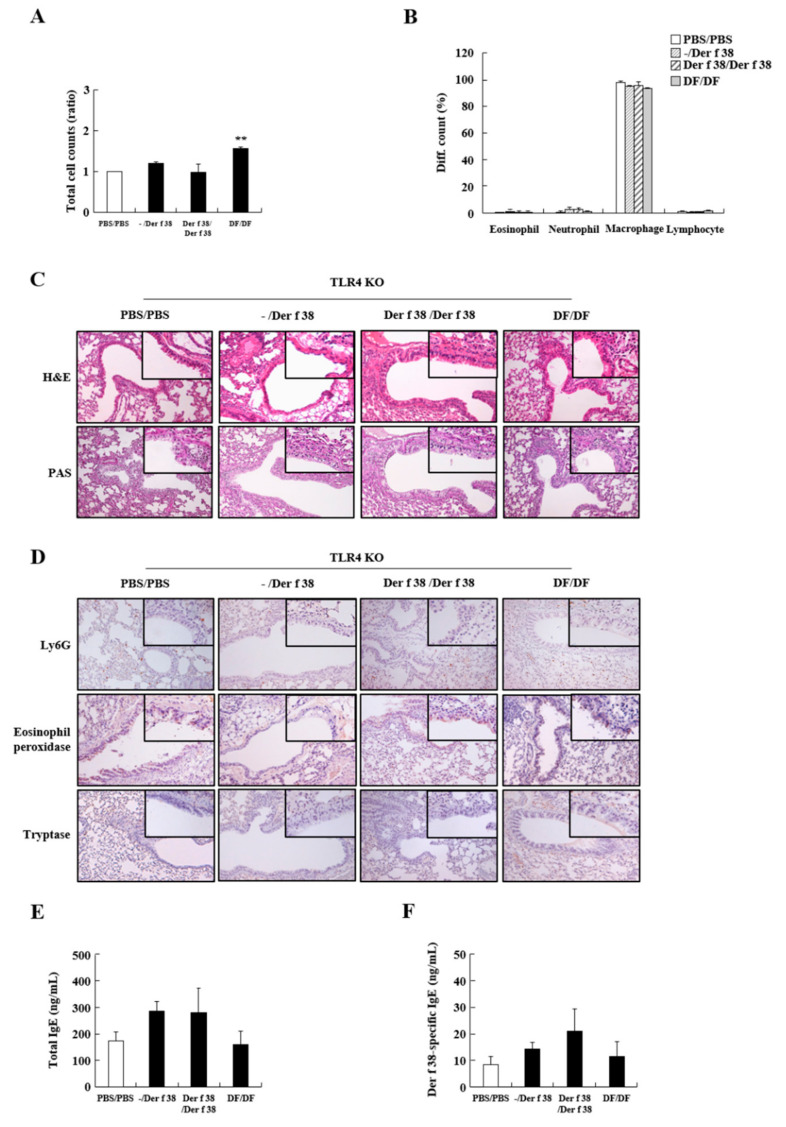
Der f 38 has no effect on asthma-like pathological alteration in TLR4 KO mice. (**A**,**B**) Total cell (**A**) and differential cell counts (**B**) in BALF of TLR4 KO mice after IN or/and IP administration of PBS, Der f 38, or DF extract. (**C**,**D**) Lung tissues from the mice were stained with hematoxylin and eosin (H&E) and periodic acid-Schiff (PAS) (**C**) or with antibodies against Ly6G, eosinophil peroxidase, and tryptase (**D**). (**E**,**F**) Total IgE (**E**) and Der f 38-specific IgE (**F**) levels in mice sera were evaluated. Data are presented as mean ± S.D. **, *p* < 0.01, a significant difference between control and stimulator-treated groups. Magnification, ×200 (**C**,**D**).

**Figure 5 ijms-22-08440-f005:**
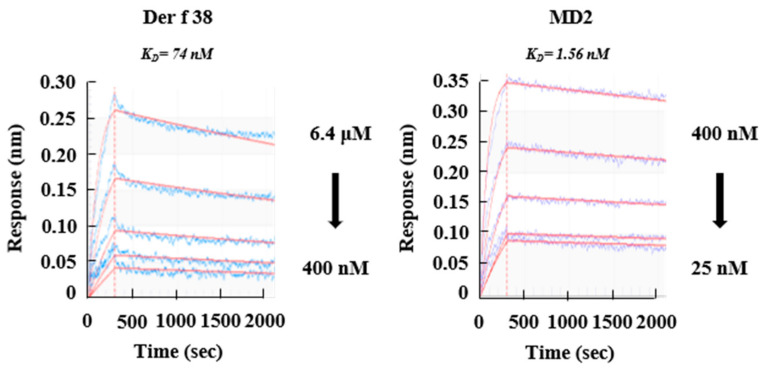
Der f 38 directly binds to TLR4. TLR4 was coated on an AR2G sensor chip. Binding affinities (*K_D_*) of Der f 38 and MD2 to TLR4 were measured by BLI as described in materials and methods.

**Figure 6 ijms-22-08440-f006:**
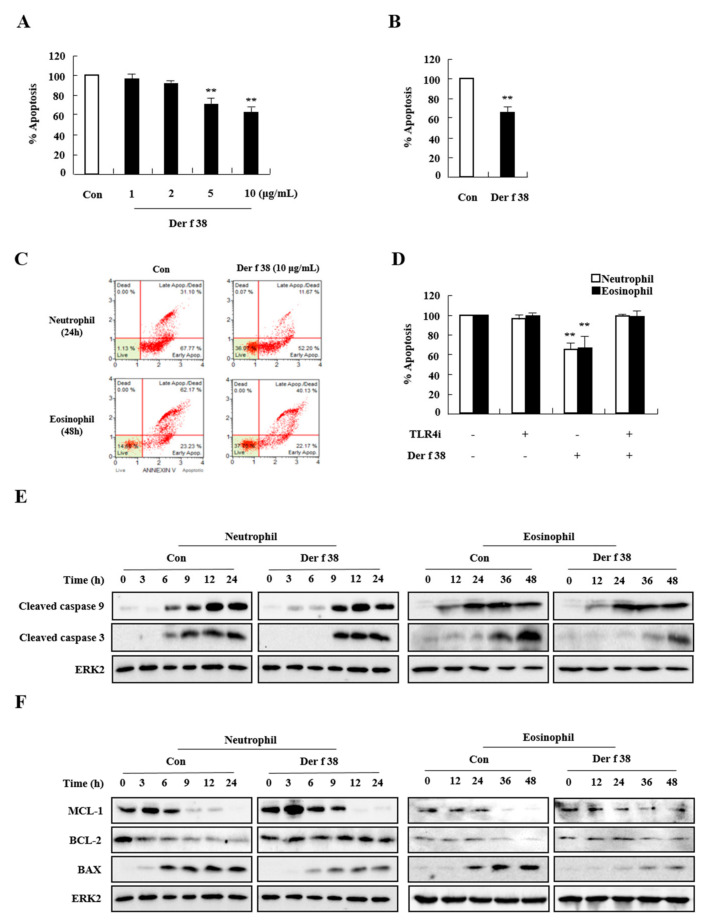
Der f 38 suppresses constitutive apoptosis of neutrophils and eosinophils. (**A-C**) Allergic neutrophils (*n* = 8) (**A**) and eosinophils (*n* = 2) (**B**) were isolated and incubated for 24 h or 48 h in the absence (Con) or presence of Der f 38. (**C**) Representative flow cytometry results derived from A and B. (**D**) Allergic neutrophils and eosinophils were pretreated with or without 2 µM TLR4 inhibitor (TLR4i) for 1 h, after which the cells were incubated for 24 h (neutrophils)or 48 h (eosinophils) in the absence or presence of Der f 38. Apoptotic cells were evaluated using annexin-PI staining. Data are presented as mean ± SD relative to the control, which was set to be 100%. ** *p* < 0.01, significant differences between control and Der f 38-treated groups. (**E**,**F**) Neutrophils and eosinophils from allergic subjects were treated with Der f 38 or without (Con) for the indicated time. Activation or expression of indicated proteins in lysates was detected by Western blotting.

**Table 1 ijms-22-08440-t001:** The results of the skin prick test.

	Normal (*n* = 20)	Allergy (*n* = 20)
DP+, DF+	6 (30%)	20 (100%)
DP+, DF−	0 (0%)	0 (0%)
DP−, DF+	2 (10%)	0 (0%)
DP−, DF−	12 (60%)	0 (0%)
Der f 38	2 (10%)	12 (60%)

## Data Availability

The data presented in this study are available in the article and Appendix A files.

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
