# Peer review of "Der f 38 Is a Novel TLR4-Binding Allergen Related to Allergy Pathogenesis from Dermatophagoides farinae"

_ijms, 2021, doi:10.3390/ijms22168440_

Round 1

Reviewer 1 Report

After correcting the huge mistake with Figures 3 and 4, the manuscript makes sense. Even more, considering the depth of investigation, logical order of the results, and the results themselves, it is an excellent contribution.

Considering that the other Reviewer (No. 3, previous submission) already provided detailed comments, I will be ascetic this time. I will have only two comments.

  1. Obligatory comments: please, go through the manuscript and identify all typos, etc., e.g. “ELSIA” in page 10 of 17. This manuscript is much too elegant in general to contain such errors.
  2. Facultative comment: please, expand the discussion a bit. For example, direct interactions between TLRs and IgE system in type 2 allergies could be discussed [PMID: 21576946 and 32883546].

Author Response

We appreciate the comments and suggestions by the reviewer. These are very helpful for improving our paper. We revised the manuscript following the reviewers’ comments and suggestions. We give an explanation on the revision by a point-to-point response to the reviewer’s comments.

1. Comment: Obligatory comments: please, go through the manuscript and identify all typos, etc., e.g. “ELSIA” in page 10 of 17. This manuscript is much too elegant in general to contain such errors.

Response: As the reviewer commented, the text has been revised as below.

Abstract: ‘TLR4and’ has been changed to ‘TLR4 and ’

Keywords: ‘House dust mouse’ has been changed to ‘House dust mite’

Figure legend: The consistent labeling format is incorporated in all figure legends.

Results: Result 2.1.

‘Figure S1’ has been changed to ‘Supplementary Figure S1’

‘(Figure 1E) (Figure S2)’ has been changed to ‘(Figure 1E and Supplementary Figure S2)’

Result 2.3.

‘Figure 3A’ has been changed to ‘Figure 3B’

‘Figure 3A-D’ has been added.

‘Figure 3E,3F’ has been changed to ‘Figure 3E,F’

‘Figure 4A,4B’ has been changed to ‘Figure 4A,B’

‘Figure 4C,4D’ has been changed to ‘Figure 4C,D’

‘Figure 4E,4F’ has been changed to ‘Figure 4E,F’

Result 2.4.

‘Figure 3’ has been changed to ‘Figure 3B,D’

‘Figure 6A and Figure S3A’ has been changed to ‘Figure 6A and Supplementary Figure S3A’

Discussion: ‘ELSIA’ has been changed to ‘ELISA’

‘Figure 3C,3D’ has been changed to ‘Figure3C,D’

‘Figure 6A and Figure S3A’ has been changed to ‘Figure 6A and Supplementary Figure S3A’

‘Figure 6B and Figure Supplementary Figure3B’ has been changed to ‘Figure 6B and Supplementary Figure S3B’

Materials and Methods: 4.6. ‘minutes’ has been changed to ‘min’

Comment 2: Facultative comment: please, expand the discussion a bit. For example, direct interactions between TLRs and IgE system in type 2 allergies could be discussed [PMID: 21576946 and 32883546].

Response: Your recommended papers are useful as explaining the relationship of TLR with allergy. However, if subtypes of TLR besides TLR4 is added in the discussion section of our manuscript, it may induce readers to be confused. We hope that you fully understand this point.

Reviewer 2 Report

In the manuscript by K. Geunyeong et. al., the authors characterize a novel allergen, Der f 38. They demonstrate that Der f 38 enhances surface 203c expression in basophils of Der f 38-positive allergic subjects and through TLR4 activation, Der f 38 increases neutrophils and eosinophils numbers by suppressing apoptosis.

This work is potentially of great interest. However, these are questions that need to be addressed.

  • Why it is important to know more about Der f 38 or new HDM allergens? I suggest adding this information in the introduction.
  • Reference 2 does not explain: “Dermatophagoides farinae (DF) is a major HDM that contains active allergic components such as arginine kinase, chitinase, cysteine protease, serine protease, fatty acid-binding proteinand peritrophin-like protein related to allergic responses”. Authors should change it for one more appropriate it.
  • Please, improve quality of figure 1A. It is blurry
  • In Figure 1E the small white letters are impossible to read. I suggest writing the amplification next to every figure or in the figure legend.
  • Figure 3A: Neither in methods nor in results DF/DF group is explained.
  • “As shown in Figure 3A, IN administration of Der f 38 enhanced neutrophil infiltration as a general effect of allergen” à It should be Figure 3B
  • It is not said that EPX stands for eosinophils peroxidase. “This phenomenon was observed when lung tissues were stained with H&E and immunologically stained with anti-Ly6G, EPX, and tryptase antibodies” Besides in Figures authors do not used EPX but say eosinophils peroxidase.
  • Figure 4 E, F: suggestion reduce Y-axis, since values are smaller. So, we can see better the difference between bars. For example, in 4E max 500ng/ml and F ma 50 ng/ml.
  • Please, change bar graphics for scatter plot with bar, or box and whiskers or scatterplot. So, we can see single values, median and mean.
  • The authors should show at least one representative flow cytometer plot for apoptosis data.
  • Figure 6 There is not F, as it said in the legend.
  • Authors should explain better how experiment in Figure S3C was performed
  • Geunyeong et. al., show how Der f 38 alters neutrophil, eosinophil, macrophage, and lymphocyte numbers (Figure 3B). These results are interesting. However, authors do not comment about lymphocytes and specially macrophages, that go in opposite direction. According to Figure 3B, allergic inflammation is linked to a decrease in the number of BALF-macrophages. Are they increasing in the tissue? Are they no protected from apoptosis as neutrophils and eosinophils? What about lymphocytes?
  • Besides, macrophage numbers are restored in the TLR4KO model, thus their reduction upon Der f 38 trigger is dependent on TLR4. Authors should investigate macrophage and lymphocyte apoptosis as they have done with neutrophils and eosinophils in figure 6. If allergic patient recruitment is not possible, at least experiments performed with healthy cells, the allergen and TLR inhibitors.
  • Geunyeong et. Al should include in the discussion the role of macrophage and lymphocytes (polarization, migration, …) in HDM-allergy according to their results and literature.

Author Response

We appreciate the comments and suggestions by the reviewer. These are very helpful for improving our paper. We revised the manuscript following the reviewers’ comments and suggestions. We give an explanation on the revision by a point-to-point response to the reviewer’s comments.